# Reduction of Viral and Bacterial Activity by Using a Self-Powered Variable-Frequency Electrical Stimulation Device

**DOI:** 10.3390/mi14020282

**Published:** 2023-01-21

**Authors:** Hsin-Yi Tsai, Yu-Hsuan Lin, Kuo-Cheng Huang, Ching-Ching Yang, Chun-Han Chou, Liang-Chieh Chao

**Affiliations:** Taiwan Instrument Research Institute, National Applied Research Laboratories, Hsinchu 300092, Taiwan

**Keywords:** electrical stimulation, viral activity inhibition, self-powered device, pulse stimulation

## Abstract

Viruses and bacteria, which can rapidly spread through droplets and saliva, can have serious effects on people’s health. Viral activity is traditionally inhibited using chemical substances, such as alcohol or bleach, or physical methods, such as thermal energy or ultraviolet-light irradiation. However, such methods cannot be used in many applications because they have certain disadvantages, such as causing eye or skin injuries. Therefore, in the present study, the electrical stimulation method is used to stimulate a virus, namely, coronavirus 229E, and two types of bacteria, namely, *Escherichia coli* and *Staphylococcus aureus*, to efficiently reduce their infectivity of healthy cells (such as the Vero E6 cell in a viral activity-inhibition experiment). The infectivity effects of the aforementioned virus and bacteria were examined under varying values of different electrical stimulation parameters, such as the stimulation current, frequency, and total stimulation time. The experimental results indicate that the activity of coronavirus 229E is considerably inhibited through direct-current pulse stimulation with a current of 25 mA and a frequency of 2 or 20 Hz. In addition, *E. coli* activity was reduced by nearly 80% in 10 s through alternating-current pulse stimulation with a current of 50 mA and a frequency of 25 Hz. Moreover, a self-powered electrical stimulation device was constructed in this study. This device consists of a solar panel and battery to generate small currents with variable frequencies, which has advantages of self-powered and variable frequencies, and the device can be utilized on desks, chairs, or elevator buttons for the inhibition of viral and bacterial activities.

## 1. Introduction

Severe acute respiratory syndrome (SARS) coronavirus 2 (SARS-CoV-2) emerged at the end of 2019. This ribonucleic acid (RNA) virus is circular and has crown-like protrusions. A study observed that SARS-CoV-2 remains viable in small droplets for a long period of time [1]. This virus rapidly spreads through droplets released while breathing, speaking, singing, coughing, or sneezing [2]; this phenomenon is known as aerosol spreading [3]. Therefore, inhibiting the activity and spread of SARS-CoV-2 has become a hot research topic. However, this virus can survive for 7–14 days in an environment with a temperature of 22 °C [4]. The survival time of SARS-CoV-2 decreases with an increase in the environment’s temperature; therefore, inhibiting the activity of this virus is also a suitable method for reducing its spread. In general, chemical and physical methods are employed to inhibit the activity of SARS-CoV-2. The adopted chemical methods include wet disinfection using alcohol, bleach, or hypochlorous acid (HOCl) water. Alcohol can dissolve the esters surrounding the mantle of SARS-CoV-2 and thus inactivate the virus. In addition, the oxidative ability of HOCl enables it to destroy the protein structure of SARS-CoV-2; thus, bleach or HOCl water can inhibit the activity of this virus.

Physical methods for inhibiting the activity of SARS-CoV-2 include dry disinfection using thermal energy, light energy, or mechanical treatment [5]. Ultraviolet (UV) C (UVC) light with a wavelength of 200–280 nm (especially 254 nm) can destroy genetic materials, such as the RNA or deoxyribonucleic acid (DNA) in human cells, as well as pathogens, such as viruses, bacteria, and fungi [6]. Moreover, thermal energy can be used to inactivate SARS-CoV-2 by aging the protein surrounding the virus. Studies have indicated that thermal energy can inactivate SARS-CoV-2 in 30, 15, and 3 min in environments with temperatures of 56 °C, 65 °C, and 95 °C, respectively [7,8]. Mechanical treatment with acoustics, microwaves, and plasma can also be employed to inactivate SARS-CoV-2. Chen et al. [9] used cold atmospheric plasma (CAP) with argon feed gas to inhibit the activity of SARS-CoV-2 on different surfaces, including metal, plastic, composite leather, and cardboard surfaces. Their results indicated that the CAP-based method is considerably safer than traditional methods, such as UV irradiation or alcohol inactivation; thus, the CAP-based method can be widely applied in the medical, scientific, and engineering fields. Yang et al. [10] used the structure-resonant energy transferred from microwaves to inactivate SARS-CoV-2. They examined the residual viral infectivity of influenza A virus under microwave illumination with different powers and frequencies, and they established a theoretical model to estimate the power threshold of microwaves for virus inactivation activity. Niknamian et al. [11] measured the frequency of SARS-CoV-2 to be 30–500 kHz by using a cyclotron. This frequency variation was caused by the variation in the virus dimension from 26 to 32 kilobases. Low-frequency magnetic fields (LFMFs) and extremely-low-frequency electromagnetic fields (ELF-EMFs) can penetrate deep tissues, cells, and mitochondria to reduce the quantity of reactive oxygen species and inflammation. Therefore, ELF-EMFs and LFMFs can be used in tandem to destroy SARS-CoV-2 in the environment and infected individuals.

Surface treatment and electric fields can also be employed to inactivate SARS-CoV-2. SARS-CoV-2 can survive on different materials for different periods of time; for example, this virus can survive for 5 days on metal surfaces, such as doorknobs; 2–3 days on plastic surfaces, such as elevator buttons; and only 4 h on copper surfaces. Cu^+^ and hydrogen peroxide can destroy the protein and ester surrounding viruses and bacteria; thus, numerous studies have used copper-based surfaces to inactivate SARS-CoV-2 [12,13,14]. Bryant et al. [15] indicated that the HuCoV-229E virus was inactivated in several minutes on a copper surface; this inactivation time was shorter than that (4 h) for HuCoV-229E in the study conducted by van Doremalen [1]. The aforementioned difference in the inactivation time was caused by the different culture media adopted in the two studies. The ClutaMAX-1 culture medium adopted in the study conducted by Bryant et al. provided higher stability and a more efficient virus inactivation effect than did glutamine culture medium used by van Doremalen. Hutasoit et al. [16] used the cold-spray technique for rapidly coating copper on steel components. A viricidal activity test indicated that 96% of the SARS-CoV-2 on steel components could be inactivated in 2 h by using this technique. The aforementioned method can be employed to stop the spread of viruses, is cheap and rapid, and can be widely applied to various components. The electrical charging of objects by using an electric field is another mechanical treatment method for inhibiting viral activity. Several studies [17,18,19] indicated that cells, bacteria, and viruses are affected by a pulsed electric current or direct current (DC) with different wavefronts. Kumagai et al. [20,21] used a constant DC potential of 1.0 V to stimulate HIV type 1 (HIV-1) and MAGIC-5 cells. Their results indicated that HIV-1LAI and HIV-1KMT infections were inhibited by 90% following 3 min of stimulation; however, the healthy cells were not damaged. These results indicate that electrical stimulation therapy is useful for the prevention of HIV-1 infection. On the basis of the preclinical evidence, Allawadhi et al. [22] proposed that the electrical stimulation method can be used to inhibit SARS-CoV-2 growth, reduce pain, boost immunity, and improve the penetration of antiviral drugs. Electric impulses damage the viral envelope by destroying the membrane and then hinder the binding of the virus to a healthy cell. Moreover, these impulses can enhance drug efficacy by damaging the viral envelope. Thus, electrical stimulation therapy can be used as a potential adjuvant for the treatment and management of COVID-19.

The aforementioned studies focused on viral activity-inhibition rates through a single process, such as mechanical treatment, thermal treatment, light irradiation, and chemical treatment. Mechanical treatment, thermal treatment, and light-irradiation methods require high power-consumption levels. Therefore, in this paper, we propose a composite method with relatively low power-consumption levels that can achieve high-efficiency viral inactivation. A cooper-based substrate was used to place the virus or bacteria sample, and an electrical stimulation module was constructed to generate small currents that flowed through the copper substrate. DC and alternating current (AC) were generated, and the magnitude and frequency of the generated AC were adjusted to investigate their effects on the viral activity. In addition, coronavirus 229E, which was used to simulate SARS-CoV-2, was placed on the copper substrate and subjected to electrical stimulation. The Vero E6 monkey kidney cell was infected by coronavirus 229E, and viral infectivity and inactivation activities were analyzed. Moreover, two common bacteria, namely, *Escherichia coli* and *Staphylococcus aureus*, were employed to test the bacteria-inhibition efficiency achieved through electrical stimulation. We determined the optimal electrical stimulation current and frequency for rapid viral and bacterial inactivation processes. The constructed electrical stimulation model can expand the application field of the electrical stimulation technique and can be used for future research on viruses and bacteria.

## 2. Fundamental Theory

The mechanism of virus and bacteria inhibition employed in this study included two areas: surface characteristics and mechanical treatment, such as an electrical field. Therefore, a metal surface, especially a copper surface, can destroy the lipid and protein surrounding a virus [12]. In addition, electrical fields, such as pulse-current or continuous-current electric fields, can generate oscillations in the microstructures and damage the structure of viruses and bacteria [20]. Therefore, we integrated the abovementioned mechanism to investigate the effects of surface characteristics and electrical fields. The virus was placed on the Cu substrate, and with the electrical stimulation in the experiment, the DNA and RNA at the virus center were released when the lipid and proteins of the virus were destroyed, and its activity was accelerated inhibited (Figure 1). The process for analyzing viral and bacterial inhibitions is described in the following text.

Generally, the inhibition rate of a virus *I_R_virus_* is defined using the logarithms of the infection values of a control and stimulation group. This rate is expressed as follows:(1)$$I_{\mathrm{R\_virus}}=\frac{10^{I_{C}}-10^{I_{S}}}{10^{I_{C}}}\times 100\%$$
where *I_C_* is the logarithm of the infection value of the control group on a blank, glass substrate, and *I_S_* is the logarithm of the infection value of the stimulation group with different stimulation parameters. The unit for *I_C_* and *I_S_* is a log plaque-forming unit (PFU)/mL. In addition, the inhibition rate of bacteria *I_R_bacteria_* is defined as follows:(2)$$I_{\mathrm{R\_bacteria}}=\frac{{\mathrm{CFU}}_{C}-{\mathrm{CFU}}_{S}}{{\mathrm{CFU}}_{C}}\times 100\%$$
where CFU_C_ is the number of bacteria on a blank, glass sample prior to electrical stimulation (the control group), and CFU_S_ is the number of bacteria on a copper tape sample following electrical stimulation performed under different stimulation parameters. CFU_C_ and CFU_S_ are expressed in colony forming units (CFUs).

## 3. Materials and Experimental Setup

### 3.1. Materials

In the experiments, copper tape was the main material used to assess viral and bacterial activity-inhibition rates. Copper tape has favorable electrical conductivity properties and can be easily pasted onto different materials. In this study, copper tape was pasted on a glass slide for passing an electrical current through the virus, namely, coronavirus 229E, and two types of bacteria, such as *E. coli* and *S. aureus*. Coronavirus 229E was used instead of SARS-CoV-2 because of the mild symptoms of infection with coronavirus 229E. Coronavirus 229E has a size of approximately 80–160 nm, and it exhibits a solar corona shape under a transmission electron microscope. This enveloped, single-stranded RNA virus causes cold symptoms when it enters the host cell and binds to a receptor. The cell infected with coronavirus 229E in this study was the Vero E6 monkey kidney cell, which is usually used for observing the Isolation and growth of the SARS coronavirus and SARS-CoV-2. To determine the relationship between the dimension, inhibition current, and frequency, *S. aureus*, which has a size of 0.5–1 μm, and *E. coli*, which has a size of 1.5–3 μm (Figure 2), was used in the activity-inhibition experiments. Suitable electrical parameters can be obtained in the future for bioengineering and activity-inhibition applications.

### 3.2. Experimental Setup

#### 3.2.1. Device and Parameters 

An electrical stimulator was designed in this study and connected to the copper tape from the bottom of the glass slide by using electrodes. The schematic of the designed stimulator and a photograph of it are presented in Figure 3 and Figure 4, respectively. The stimulation type or frequency of a traditional pulse generator was fixed in the stimulation process, and the power consumption was high. The novelty of the designed electrical stimulator included variable frequency, alternation in DC and AC stimulations in the stimulation process, and self-powered electrical stimulation transferred by a solar panel and battery. It has advantages of wide applicability for different viruses and bacteria, low power consumption, and no requirement for external power. Additionally, this electrical stimulator can be integrated into a rigid or flexible substrate for flat- or arc-shaped-object applications in the future. The length and thickness of the glass substrate used in the experiment were 75 and 1 mm, respectively, and the distance between the electrode at the backside of glass substrate and edge was 4 mm. Therefore, the distance between the tested sample (virus and bacteria) and electrical stimulation electrode was approximately 42 mm. The operating voltage of the stimulator was 5 V, and the current, pulse width of the stimulation, and number of stimulation counts could be adjusted.

The stimulation current of the designed device can be adjusted between 1 and 50 mA at intervals of 1 mA (Figure 5a), and the pulse width can be set as 40, 50, 100, 200, 500, and 1000 ms per pulse (Figure 5b). Different pulse widths indicate different frequencies of electrical stimulation. In addition, the number of stimulation counts can be set as 5, 10, 25, 50, 100, and 125. Pulse stimulation with a certain pulse width can be divided into two types: DC (zero-to-positive signal) (Figure 5b) and AC (positive-to-negative signal) pulse stimulations (Figure 5c). These two types of stimulation might cause different electrical resonances for viruses or bacteria. In the experiments conducted in this study, the pulse width and stimulation counts were initially fixed, and the stimulation current was varied to identify a suitable current. After a suitable current was obtained, the pulse width and stimulation count were varied to investigate their effects on viral or bacterial activity-inhibition rates. In this investigation, the total stimulation time was fixed at 10 s. In addition, continuous-current stimulation was performed to inhibit bacterial and viral activities. The experimental parameters (i.e., stimulation current, pulse width related to frequency, and stimulation counts) are presented in Table 1.

#### 3.2.2. Analysis of Viral Activity Inhibition

Virus quantification involves counting the number of viruses in a specific volume to determine the virus concentration value. This method can be used for R&D and production in commercial and academic laboratories, where the virus quantity during each process is a crucial parameter. In 1952, Dulbecco [23] applied phage plaque technology to animal virology, thus adopting the plaque assay as a virus quantification method. The viral plaque assay is a standard method used for determining virus concentrations based on the infectious dose, which refers to the number of PFUs in a virus sample. This number is determined by using microbiological methods in cell culture dishes or multiwell plates. Specifically, a cell culture dish is filled with a monolayer of host cells, and the virus is then diluted to different concentrations. Subsequently, the monolayer of host cells is infected with the diluted virus and treated with a semisolid medium (such as agar or carboxymethyl cellulose) to cover the virus and infected cells to prevent the viral infection from spreading indiscriminately. When the cells are infected by the virus, they lyse and spread the virus to neighboring cells, where the cycle of infection–cell lysing is repeated. The infected area of the cells then forms a plaque, which can be observed through optical microscopy or with the naked eye. Subsequently, the semisolid medium is poured out from the multiwell plates, after which crystal violet solution is added to the dish for 15 min until it stains the cell cytoplasm. Excess water from the plates is then gently removed, and the remaining dead cells are colorless. The formation of empty plaques may require 3–14 days, depending on the virus, and the viral plaques are typically counted manually. Each viral plaque formed is assumed to represent one infectious viral particle, and the number of infectious viral particles per milliliter is determined. To quantify the viral titers, only plates containing 10–100 viral plaques are counted. Viral sample titers are quantified when 100 viral plaques are counted, and a ± 10% variation is observed in the number of viral plaques. When the number of PFUs or CFUs is reduced by 10 times, the reduction is defined as a 1-log reduction (log1) and represents 90% disinfection and sterilization. A decrease in the number of viruses or bacteria on the surface of a test object by 4-log from 1,000,000 to 100 CFU or PFU indicates the potential disinfection of 99.99% of harmful microorganisms. Figure 6 displays the viral plaques formed in this study by coronavirus 229E on cell monolayers. The cells displayed in this image were stained with crystal violet to form white, empty viral plaques where the virus infected the destroyed monolayer of the host cells.

In the viral activity-inhibition experiments, the Vero E6 cell and coronavirus 229E, which has low pathogenicity, were used, and the testing environment was a biosafety level 2 laboratory (BSL-2 Lab). The developed device was placed in the BSL-2 Lab, and the virus was placed on clear copper tape and electrically stimulated under different electric stimulation parameters. The evaluation of bacterial activity inhibition is similar to that of viral activity inhibition, and bacterial activity-inhibition performance was evaluated by comparing the numbers of viable cells in the control and stimulation groups.

#### 3.2.3. Analysis of Bacterial Activity Inhibition

For the analysis of bacteria activity inhibition, 100 μL of bacterial solution (containing 10^7^ CFU of bacteria) was placed on a glass substrate so that approximately 10^6^ CFUs of bacteria were present on the substrate. The substrate was then electrically stimulated under certain parameters for 10 s. Following electrical stimulation, the sample on the substrate was extracted through serial dilution by using 5 mL of phosphate-buffered saline. A total of 1 mL of each dilution was placed in a Petri dish, to which 15–20 mL of tryptone soy agar was added. Finally, the bacteria and cells were incubated at 32.5 ± 2.5 °C for 3 days, and the bacterial concentration following stimulation was analyzed. The abovementioned experiments were operated by the professional operator in the standard laboratory of the SGS Taiwan company, and the process was performed according to the standard: USP 51 Antimicrobial Effectiveness test. 

#### 3.2.4. Experimental Process

Three primary steps were involved in developing the electrical stimulation device and using it to obtain the viral and bacterial activity-inhibition rates. These steps are described as follows:The circuit and structure of the designed electrical stimulation device were developed using three-dimensional printing technology. The current, pulse width, and stimulation count could be adjusted when using this device (Table 1). A piece of copper tape on a glass-slide cover were used as the testing and conducting substrates.The cell, virus, and bacteria samples were prepared, and the virus or bacteria was placed on the copper tape for electrical stimulation. Initially, the current was varied under a fixed pulse width and stimulation count to determine the suitable currents for the virus and bacteria. Details on cell preparation, virus amplification, and the adopted antiviral assay are provided in Section 3.2.2 and Section 3.2.3.After the suitable currents related to the virus and bacteria were determined, the current was fixed, and the pulse width and stimulation count were adjusted to determine their influences on bacterial and viral activity-inhibition rates. Each experiment was performed at least twice, and the mean values were considered in the analysis.

## 4. Experimental Results and Discussion

To investigate the effect of electrical stimulation on the relevant virus and bacteria samples, suitable stimulation parameters were sequentially investigated in this study. The virus and bacteria were tested separately using the same device to eliminate the experimental variations caused by the device. After suitable stimulation currents were determined for the virus and bacteria, the effect of the pulse width on the bacterial and viral activities was examined. The experimental results are presented in the following sections.

### 4.1. Inhibition of Viral and Bacterial Activities on the Metal Plate

The virus and bacteria were placed on the blank, glass substrate and copper tape to investigate the viral and bacterial activity-inhibition rates caused by copper ions. The viral and bacterial infection values on the blank, glass substrate were used as the control values, and the viral and bacterial activity-inhibition rates achieved with the copper tape were calculated using Equation (1). The viral activity-inhibition rate obtained with the copper tape was approximately 12.9% (Table 2). In addition, the inhibition rates of *E. coli* and *S. aureus* activities obtained with the copper tape were less than 1% and 3.3%, respectively, because proteins do not surround *E. coli* and *S. aureus*; thus, their activities are not considerably inhibited by copper ions. The results indicate that the copper tape causes a marginal reduction in the viral activity, but negligibly affects the bacterial activity. Electrical stimulation was subsequently performed to examine whether electrical stimulation affected the viral and bacterial activity-inhibition rates.

### 4.2. Effect of Stimulation Current on Virus and Bacteria

After the effects of applying a copper tape on the virus and bacteria were investigated, the virus and bacteria were subjected to different stimulation currents through the copper tape under a fixed pulse width and stimulation count of 100 ms and 10 counts, respectively. The results indicate that the viral activity-inhibition rate increases with an increase in the stimulation current, where the inhibition rate was 2.27% under a 5 mA stimulation current and increased to 16.82% under a 25 mA stimulation current. It illustrated that the higher current had a greater impact on the virus activity-inhibition rate. Additionally, the inhibition effect was saturated when the stimulation current was higher than 25 mA (Figure 7). The results indicate that an increase in the stimulation current can enhance the viral activity-inhibition rate and it has a limitation; hence, the other stimulation parameters should be adjusted to enhance the inhibition rate. Because the activity-inhibition effect saturated when the stimulation current reached 25 mA, a stimulation current of 25 mA was adopted for the virus in the following experiments to avoid excess power consumption.

The inhibition rates of *E. coli* and *S. aureus* activities were tested using the same device. The results indicate that the inhibition rate of *E. coli* activity achieved with the developed device was lower than 10% when the stimulation current was less than 25 mA, and this value increased to 10.7% when the stimulation current was increased to 50 mA (Figure 8). Moreover, the inhibition rate of *S. aureus* activity increased from 19.7% to 26.2% when the stimulation current was increased from 5 to 50 mA. The results indicate that under the same stimulation current, a lower activity-inhibition rate is achieved for larger bacteria; thus, to achieve the same activity-inhibition rate, a higher stimulation current is required for *E. coli* than for *S. aureus* because *E. coli* is larger than *S. aureus*. In addition, the inhibition rate of *S. aureus* activity was four times that of *E. coli* activity under a low stimulation current of 5 mA. Moreover, the benefit (in terms of reducing activity) of increasing the current from 5 to 50 mA was considerably smaller for *S. aureus* than for *E. coli*. To save power, the stimulation currents for inhibiting *E. coli* and *S. aureus* activities were set as 50 and 5 mA, respectively, in the following experiments. These values represent the lowest currents at which an activity-inhibition rate of 10% was achieved.

### 4.3. Effects of Stimulation Type and Frequency on the Inhibition of Viral Activity

The pulse stimulation type and pulse width were adjusted to investigate their effects on viral and bacterial activity-inhibition rates. In this investigation, the total stimulation time was fixed at 10 s. The experimental results indicate that the activity-inhibition rates achieved with pulse stimulation are generally higher than those achieved with continuous-current stimulation. The viral activity-inhibition rate was higher than 60% under DC pulse stimulation conditions with frequencies of 2 and 20 Hz, and less than 40% under DC pulse stimulation conditions with other frequencies (Figure 9). The viral activity-inhibition rate under DC continuous stimulation conditions was approximately 47%. Moreover, this rate was less than 40% under AC pulse stimulation conditions. The inhibition rate had a maximum value of 33% under AC pulse stimulation conditions with a frequency of 5–20 Hz and a value of approximately 10% under AC continuous stimulation conditions. According to the above results, the DC pulse stimulation with specific frequency such as 2 or 20 Hz may have the resonance or oscillation with the virus, and it speed up the damage of the virus lipid and protein and enhance the inhibition rate. In addition, the viral activity-inhibition rate was lower under AC pulse stimulation than under DC pulse stimulation because compared with positive-zero electrical stimulation cycles, positive–negative electrical stimulation cycles reduce the effect of current stimulation in decreasing the viral activity-inhibition rate. Therefore, DC pulse stimulation at frequency rates of 2 and 20 Hz was the best stimulation condition for reducing the viral activity-inhibition rate. 

### 4.4. Effect of the Pulse Width of AC Pulse Stimulation on the Inhibition of Bacterial Activity

Under a fixed stimulation current of 50 mA, the pulse width was adjusted to investigate its effect on the inhibition rate of *E. coli* activity. An inhibition rate of 17.5% was achieved with DC and AC continuous simulation conditions. The inhibition rate of *E. coli* activity varied marginally at frequencies less than 5 Hz. In particular, the inhibition rate at 2 Hz was different to those at 1 and 5 Hz under DC and AC stimulation conditions (Figure 10). The inhibition rate decreased and increased with an increase in the stimulation frequency beyond 5 Hz under DC and AC pulse stimulation conditions, respectively. The results indicate that AC pulse stimulation has a superior inhibitory effect on *E. coli* activity than DC pulse stimulation conditions. In particular, the inhibition rate reached approximately 80% at a stimulation frequency of 25 Hz under AC stimulation conditions.

The inhibition rate of *S. aureus* activity is higher than that of *E. coli* activity under the same current because *S. aureus* is smaller than is *E. coli*. Therefore, the stimulation current was fixed as 5 mA in the experiments conducted on *S. aureus*. The results indicated that the inhibition rate of *S. aureus* activity was marginally higher under continuous AC stimulation than under continuous DC stimulation. However, the inhibition rate of *S. aureus* activity value was higher under DC pulse stimulation than under AC pulse stimulation. In addition, the trend of the inhibition rate with the stimulation frequency was the same under DC and AC pulse stimulation (Figure 11). The best stimulation frequencies for inhibiting *S. aureus* activity were revealed to be 2 and 10 Hz.

From the above results, the inhibition rate of *E. coli* and *S. aureus* showed that the bacteria with larger dimension needs the higher pulse frequency and electrical field alternation to generate the oscillation with electrical field and let the structure damage. Therefore, the stimulation frequency was a critical parameter for virus and bacteria inhibition. 

## 5. Conclusions

In this study, we developed an electrical stimulation device to reduce viral and bacterial activities and investigated the effects of the stimulation current, stimulation type (DC or AC continuous or pulse stimulation), and stimulation frequency on the activity-inhibition rates for one virus (coronavirus 229E) and two types of bacteria (*E. coli* and *S. aureus*). Even though we cannot really confirm the mechanism of virus and bacteria inhibition, but we conjectured that the lipids and proteins around the virus react with the copper material and destroy its structure and the pulse electrical stimulation with specific can accelerate the activity-inhibition rate. The experimental results verify our ideas and reveal that the inhibition rate of coronavirus 229E increases with an increase in the stimulation current and is saturated when the current reaches 25 mA. The inhibition rate of coronavirus 229E activity reached 70% in 10 s under DC pulse stimulation conditions with a frequency of 2 or 20 Hz. In addition, to achieve the same activity-inhibition rate, a higher stimulation current was required for *E. coli* (the larger bacteria) than for *S. aureus*. A superior activity-inhibition effect was achieved for *E. coli* under the AC pulse stimulation conditions with a frequency of 25 Hz, with the activity-inhibition rate reaching approximately 80%. The inhibition trend of *S. aureus* activity with different frequencies was the same under DC and AC pulse stimulation. This device was a self-powered stimulation device with a solar battery to save the power in electrical stimulation process. This system collects light from the environment and converts it into electrical energy. The results of this study indicate that the developed device can generate variable-frequency stimulation for inhibiting the activities of coronavirus 229E, *E. coli*, and *S. aureus*. This system does not require external power, does not cause human injury, and has high compatibility with numerous devices, such as student’s desks and chairs, or elevator buttons that are commonly used in daily life. The developed system can be used to reduce the infection rates and spread of viruses and bacteria, and the results of this study can act as a reference for reducing viral and bacterial infection rates in biomedical and health applications. In the future, the protozoa, such as plasmodium or toxoplasma, and other worms can be used to replace the virus and bacteria for electrical stimulation and investigate its activity-inhibition rate on humans or cells for novel innovations and applications. 

## Figures and Tables

**Figure 1 micromachines-14-00282-f001:**
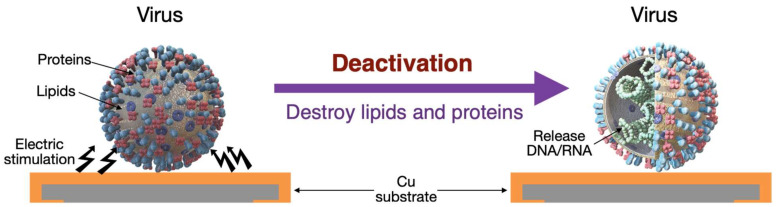
Schematic of a virus deactivated by a copper substrate through electrical stimulation.

**Figure 2 micromachines-14-00282-f002:**
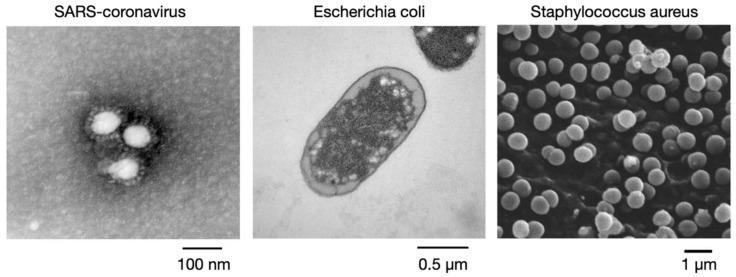
Schematic of the tested virus and bacteria (Source: EM Atlas of Clinical Microbes, Taiwan Centers for Disease Control).

**Figure 3 micromachines-14-00282-f003:**
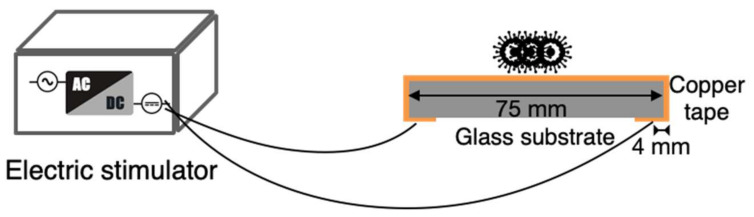
Schematic of the designed electrical stimulation device connected to a copper tape on a glass substrate.

**Figure 4 micromachines-14-00282-f004:**
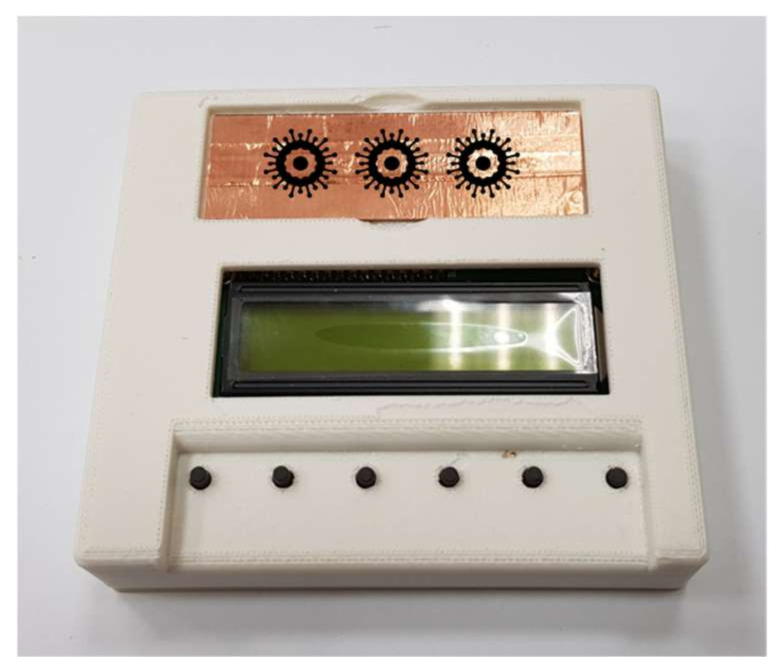
Photograph of the designed electrical stimulation device.

**Figure 5 micromachines-14-00282-f005:**
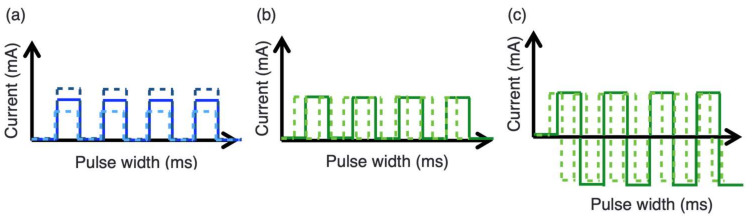
Wave diagrams of (**a**) different stimulation currents, (**b**) direct-current (DC) pulse stimulation, and (**c**) alternating-current (AC) pulse stimulation with different pulse widths.

**Figure 6 micromachines-14-00282-f006:**
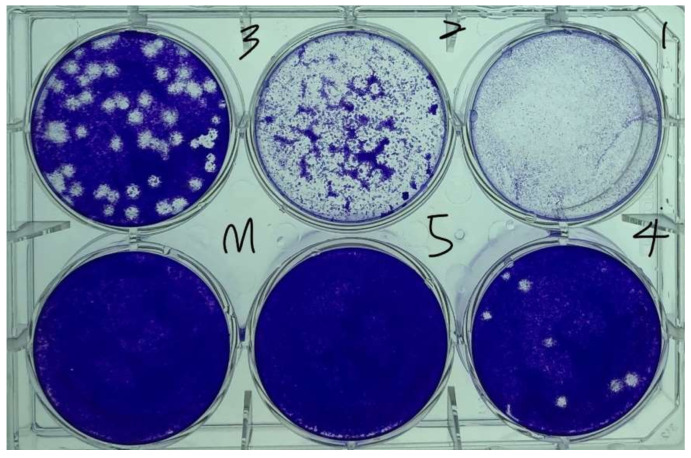
Cell and virus plaques in a virus plaque assay (samples 1–5 were obtained through a 10-fold dilution of the original virus solution).

**Figure 7 micromachines-14-00282-f007:**
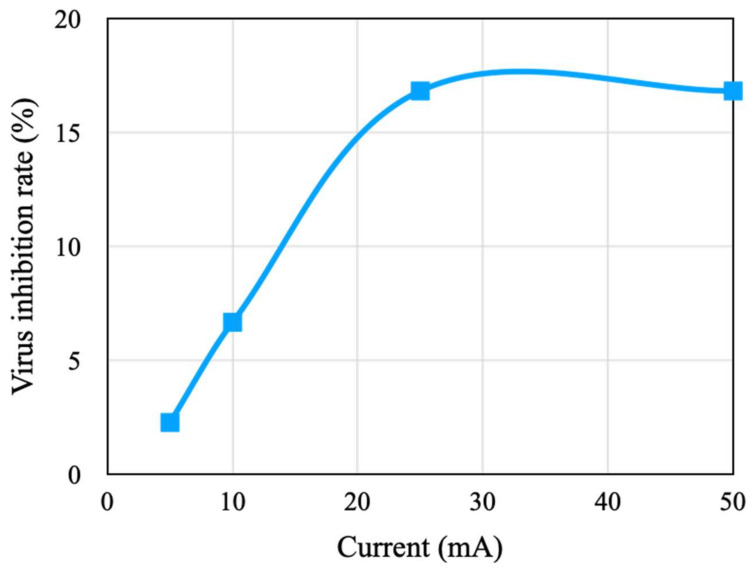
Viral activity-inhibition rates under different stimulation currents.

**Figure 8 micromachines-14-00282-f008:**
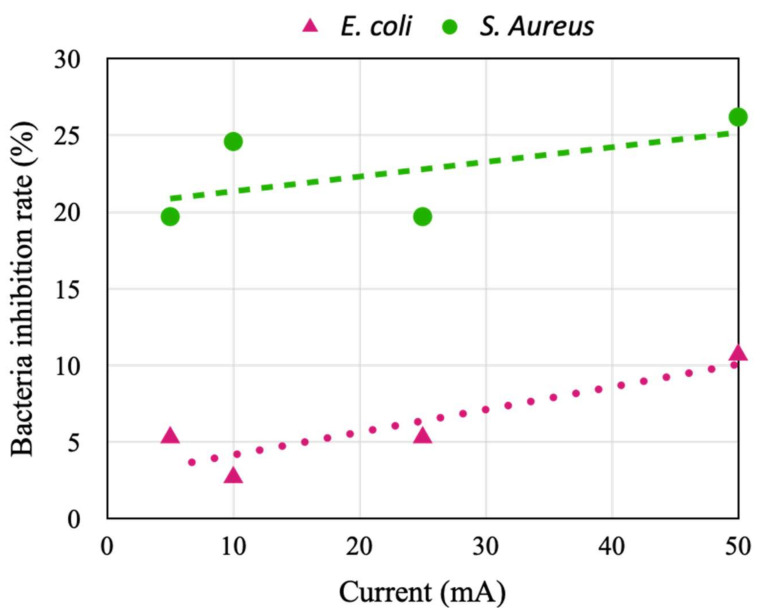
Inhibition rates of *Escherichia coli* and *Staphylococcus aureus* activities under different stimulation currents.

**Figure 9 micromachines-14-00282-f009:**
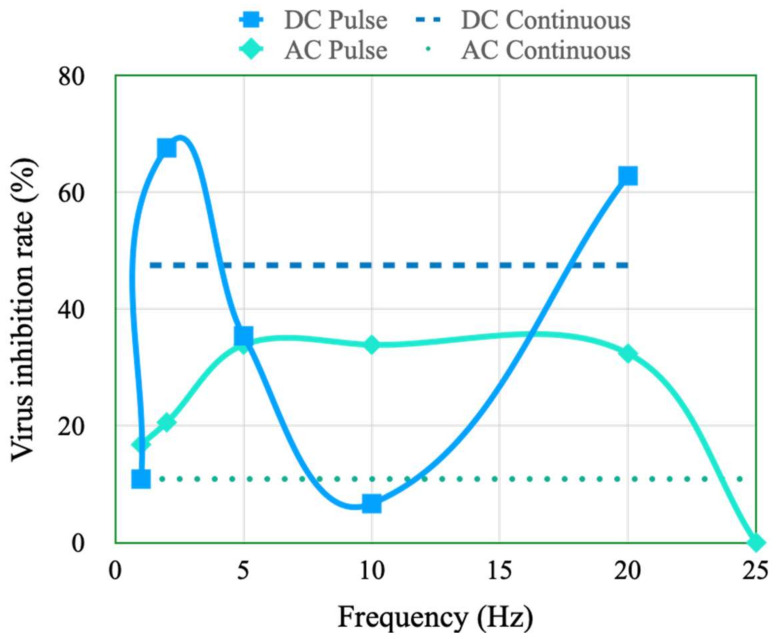
Viral activity-inhibition rates under DC and AC continuous and pulse stimulation conditions with different frequencies.

**Figure 10 micromachines-14-00282-f010:**
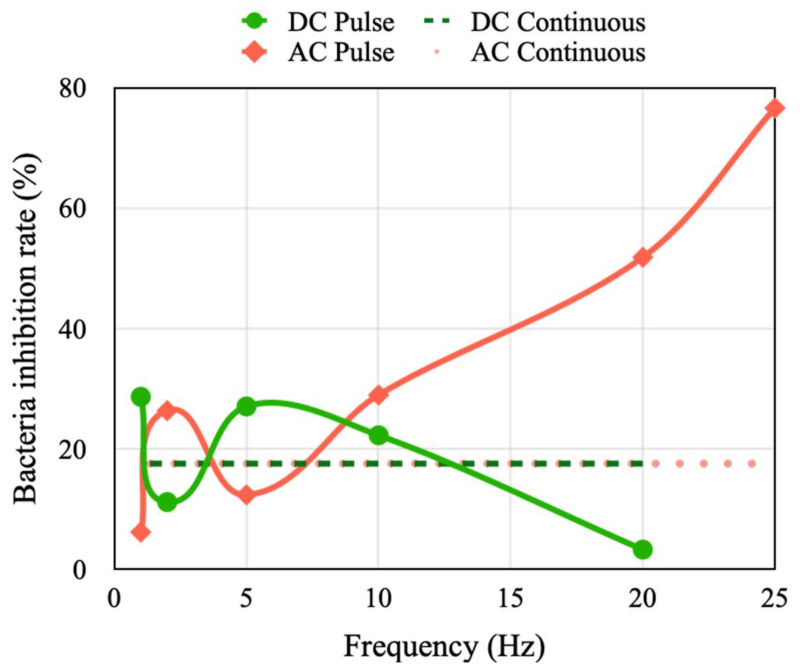
Inhibition rate of *E. coli* activity under DC and AC continuous and pulse stimulation conditions with different frequencies.

**Figure 11 micromachines-14-00282-f011:**
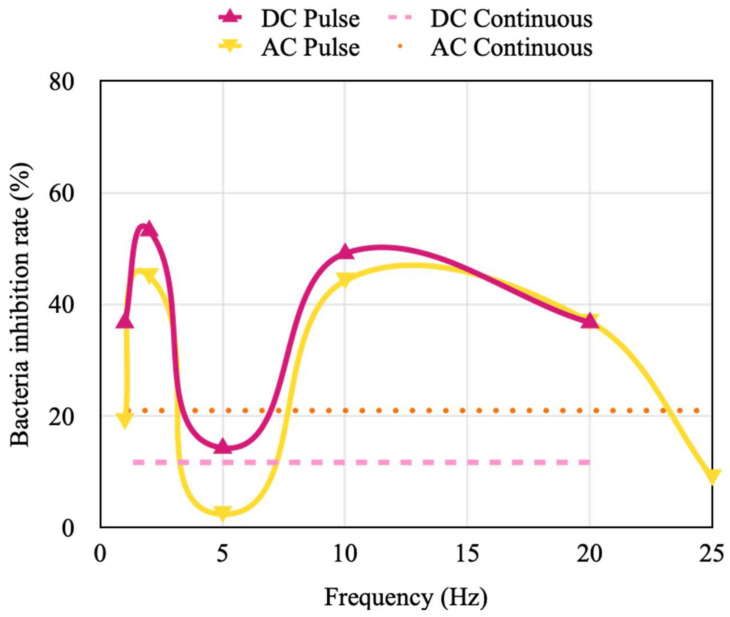
Inhibition rate of *S. aureus* activity under DC and AC continuous and pulse stimulation with different frequencies.

**Table 1 micromachines-14-00282-t001:** Stimulation current, pulse width, and stimulation count in the conducted experiments.

Current (mA)	Pulse Width (ms)	Frequency (Hz)	Count (Time)
5	Continuous		
	40	25	125
10	50	20	100
	100	10	50
25	200	5	25
	500	2	10
50	1000	1	5

**Table 2 micromachines-14-00282-t002:** Viral and bacterial activity-inhibition rates achieved with the copper plate without electrical stimulation.

Virus/Bacteria	Coronavirus 229E	*E. coli*	*S. aureus*
Inhibition rate (%)	12.9	<1	3.3

## Data Availability

Data is contained within the article.

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
