# Peer review of "Reduction of Viral and Bacterial Activity by Using a Self-Powered Variable-Frequency Electrical Stimulation Device"

_micromachines, 2023, doi:10.3390/mi14020282_

Round 1
Reviewer 1 Report
The studies that have been conducted by Yi-Tsai et all are quite good and have provided information about innovative tools or methods, especially the electric stimulation method to stimulate (inhibited) the virus and bacteria. However, I need to confirm/clarify some of the points that have been written in this article.
1. When writing scientific names, such as E. coli, Staphylococcus aureus, and others, I suggest it is preferable to use italics.
2. In lines 12-13, the authors wrote: ..... then investigate the infectivity on healthy cells. Could you please clarify these points, I could not find any information on how you performed or whether the electrical pulse has a good or bad effect on the healthy cells.
3. Furthermore, the distance used in this study when electrical stimulation is applied to viruses and bacteria is critical (I mean the optimized distance).
4. My recommendation is that in future studies, protozoa like (plasmodium, toxoplasma, etc.) may need to be inhibited in order to introduce new innovations.
Author Response
Reviewer#1
The studies that have been conducted by Yi-Tsai et all are quite good and have provided information about innovative tools or methods, especially the electric stimulation method to stimulate (inhibited) the virus and bacteria. However, I need to confirm/clarify some of the points that have been written in this article.
Response: Thanks for the reviewer’s suggestion. In the revised paper, several points were confirmed, clarified and re-written. In addition, the English were edited by professional English Editing Office.
- When writing scientific names, such as E. coli, Staphylococcus aureus, and others, I suggest it is preferable to use italics.
Response: Thanks for the reviewer’s suggestion, the names of virus and bacterial include Escherichia coli and Staphylococcus aureus were written as italics in the revised manuscript.
- In lines 12-13, the authors wrote: ..... then investigate the infectivity on healthy cells. Could you please clarify these points, I could not find any information on how you performed or whether the electrical pulse has a good or bad effect on the healthy cells.
Response: Thanks for the reviewer’s suggestion. In line 12-13, the healthy cell indicated the cell in the infection process by virus or bacteria after electric stimulation instead of the direct effect of electric stimulation on healthy cell. For example, there was Vero E6 monkey kidney cell in virus infection experiment in Section 3.1 in the revised manuscript.
- Furthermore, the distance used in this study when electrical stimulation is applied to viruses and bacteria is critical (I mean the optimized distance).
Response: Thanks for the reviewer’s suggestion. We assumed that the electrical conductivity of the copper tape is consistent in the experiment. The distance between the electric stimulation electrode and the tested sample (virus and bacteria) was approximately 42 mm in this study. The detail was described in the Section 3.2.1 in the revised manuscript. In future research, the electrical stimulation distance between the electrode and tested sample can be optimized for enhancing the inhibition rate.
- My recommendation is that in future studies, protozoa like (plasmodium, toxoplasma, etc.) may need to be inhibited in order to introduce new innovations.
Response: Thanks for the reviewer’s suggestion. In the future, we can use the protozoa like the plasmodium or toxoplasma, or other worms as the tested sample for electric stimulation and investigate it activity inhibition on human or cells for new innovations and applications. The description was written in the Conclusion in the revised manuscript.
Please see the attachment.

Reviewer 2 Report
I do not understand what the authors trying to say. What is the main topic of your article? Instrument or method? Where is the originality of research?
Author Response
I do not understand what the authors trying to say. What is the main topic of your article? Instrument or method? Where is the originality of research?
Response: Thanks for the reviewer’s suggestion. In the revised paper, we modified the main topic of the article and highlight in the Abstract and last paragraph of the Introduction. Therein, the traditional viral activity inhibition usually through single process, such mechanical treatment, thermal treatment, light irradiation, and chemical treatment. Therefore, we propose a composite method included copper substrate and electric stimulation to achieve the high efficiency viral and bacterial inactivation and present the optimal electric stimulation parameters like current and frequency for different types of viral and bacterial. This device and experimental result can provide the reference information for inhibition of viral and bacterial activity and can be employed ton desks, chairs, or elevator buttons. (Main topic and originality of research).
In addition, the instrument and method were described in Section 3.2. The electric stimulation device was self-designed and conducted to provide the AC, DC pulse and continuous stimulation, and stimulation parameters like electric current and pulse width can be adjusted. The tested sample (virus or bacteria) were placed on the copper substrate on the electric stimulation device and then stimulated with different parameters. In addition, the tested sample was placed into the dished to infect the cells, and then analyzed the inhibition rate between the control group and experimental group.
Please see the attachment.

Round 2
Reviewer 2 Report
The manuscript was improved a little bit, but still lack of clearly represented idea and novelty. 1) the title of manuscript is not correct and misleading; 2) introduction is over-crowded, please reduce the size and extract the most important information which will be covered by the experiments; 3) Figure 3. the information provided here is not covered by your experiment, what is the meaning of Cu^29? The destruction of lipid and proteins was not the scope of your experiments; 4) the "electric stimulator" itself there is a regular commercially available pulse generator, where is the novelty here? 5) the analysis of viral activity is not new, the methods must be cited; 6) the table 2 is useless; 7) Chapter 3.2.3. must be done according to the standard, like ISO 18593:2004 or others; 8) Fig. 7 does not have statistical analysis; 9) Fig. 8 - not statical analysis as well; 10) the effects of frequency to the viral and bacterial inhibition rate must be explained. 11) the mechanism of inhibition is not revealed.
Author Response
Reviewer’s comment,
Reviewer#2
The manuscript was improved a little bit, but still lack of clearly represented idea and novelty.
Response: Thanks for the reviewer’s suggestion. In the revised paper, we modified the main topic of the manuscript, and highlight the important information such as the mechanism related the experiment and results. The innovations of this manuscript are: 1. Self-powered, which means that activity of bacteria and viruses can be inhibited without electric wires, and this advantage helps the technology to spread to many places. 2. Variable-frequency electrical stimulation, we try to observe the impact on biological activity under different electrical frequency stimulation conditions. So far, no related research has been seen in the previous literature. This device and experimental result can provide the reference information for inhibition of viral and bacterial activity and can be employed to different applications. We expect to improve public health through this new technology.
- the title of manuscript is not correct and misleading;
Response: Thanks for the reviewer’s suggestion. We modified the title of the manuscript and focused on the experiment topic from “Inhibition of Viral and Bacterial Activity Through Electrical Stimulation” to “Reduction of Viral and Bacterial Activity by Using a Self-powered Variable-frequency Electric Stimulation Device”. Please find the revised manuscript.
- Introduction is over-crowded, please reduce the size and extract the most important information which will be covered by the experiments;
Response: Thanks for the reviewer’s suggestion. We removed some reference and kept the important reference such as the related studies of virus inhibition method by thermal energy, light energy and mechanical treatment. Therein, the mechanical treatment included the atmospheric plasma, resonant energy, surface treatment and electric field for the experiment. We marked the sections that are expected to be removed, please find the Introduction in the revised paper. Additionally, we will modify the reference number if the modification is suitable.
- Figure 3. the information provided here is not covered by your experiment, what is the meaning of Cu^29? The destruction of lipid and proteins was not the scope of your experiments;
Response: Thanks for the reviewer’s suggestion. The 29 is the atomic number, and we have removed it in the Figure 1 in the revised manuscript. In addition, this experiment didn’t include the destruction of lipid and protein around, and we assumed the lipid membrane and protein was destroyed by the Cu substrate from the previous research, and the electric stimulation will accelerate the destruction of structures. We have modified the description in Section 2 in the revised manuscript.
- the "electric stimulator" itself there is a regular commercially available pulse generator, where is the novelty here?
Response: Thanks for the reviewer’s suggestion.
Traditionally, the electrical stimulation achieved by commercial (traditional) pulse generator usually has a fixed frequency or pulse type (AC or DC). This method is rigid (not smart), so it consumes more power. That is, it requires wires to constantly provide enough power. The innovation of this study is to provide the frequency and pulse type changes during electrical stimulation. These actions can improve the inhibition efficiency of viruses and bacteria, so that the power used becomes less.
We had added the solar panel and battery to collect, transfer and storage the power in the designed electric stimulator, and it can provide tiny (enough) current without external power. This module also can be integrated into the flexible substrate and can be applied to the arc-shaped object for activity inhibition of different virus and bacteria. Please find the description about the novelty in Section 3.2.1 in the revised manuscript.
- the analysis of viral activity is not new, the methods must be cited;
Response: Thanks for the reviewer’s suggestion. The analysis of viral activity was presented by Dulbecco in 1952, we highlighted the citation in the Section 3.2.2 in the revised manuscript.
- the table 2 is useless;
Response: Thanks for the reviewer’s suggestion. In the revised paper, we removed the original table 2 in Section 3.2.2.
- Chapter 3.2.3. must be done according to the standard, like ISO 18593:2004 or others;
Response: Thanks for the reviewer’s suggestion. The analysis of bacterial activity inhibition was according to the standard:USP 51 Antimicrobial Effectiveness Test, and the experiment was operated by the standard laboratory of SGS Taiwan. The information was added in the Section 3.2.3 in the revised manuscript.
- Fig. 7 does not have statistical analysis;
Response: Thanks for the reviewer’s suggestion.
Since the cost of experiments exceeded our research budget, we reduced the number of samplings. Therefore, it is difficult to accurately calculate the P value (for statistical analysis). Even so, the experimental results still be reliable and have considerable reference value. Each measurement result is averaged from at least 2 sets of data, and the experimental procedure is in line with USP51 testing standards (SGS laboratory). We expect you to understand our difficulties with limited research funding. This study mainly expects to observe the correlation and trend, and does not intend to accurately quantify the correlation curve. In the revised paper, we added some analysis and description for Fig. 7. Therein, the virus inhibition rate was 2.27 % under 5 mA stimulation and the inhibition rate increased to 16.82 % under 25 mA stimulation. It indicated that the higher current had the larger impact on virus, but it had a limitation and the inhibition rate saturated when the stimulation current was higher than 25 mA. Therefore, the other stimulation parameters should be adjusted to enhance the inhibition rate. Please find the Section 4.2 in the revised manuscript.
- Fig. 8 - not statical analysis as well;
Response: Thanks for the reviewer’s suggestion.
Due to the insufficient number of samples, the P value still could not be accurately obtained. The reason is the same as above, please understand. This study mainly expects to verify the efficacy of the developed module and method, and does not intend to accurately quantify the correlation curve. The experimental results indeed verified our idea: different electrical stimulation frequencies and pulse patterns have different inhibition rates for viruses and bacteria. The results are valuable and will contribute to the improvement of energy conservation and public health in the future. In the next stage, we will invest in more related research to establish a more accurate regression curve.
- the effects of frequency to the viral and bacterial inhibition rate must be explained.
Response: Thanks for the reviewer’s suggestion. Owing to the dimensional and surface property, the DC and AC stimulation with different frequency had different effects on activity inhibition. For virus, the frequency of 2 or 20 Hz may have resonance or oscillation with virus, and the oscillation was too small to damage the structure and cause the smaller inhibition rate in other frequencies. For bacteria, the inhibition rate was affected by the frequency and bacteria’s dimension, which the bacteria with larger dimension need the higher stimulation frequency to obtain the better inhibition rate. The caused reason was the higher frequency and quickly alternation of electric field can generate the oscillation and let the structure damage for large dimension of bacteria like E. coli. Please find the description in Section 4.3 and 4.4 in the revised paper.
- the mechanism of inhibition is not revealed.
Response: Thanks for the reviewer’s suggestion.
In fact, we cannot really confirm the inhibition mechanism of viruses and bacteria, but we conjectured that the lipids and proteins around the virus react with the copper material and destroy its structure. In addition, electrical stimulation can produce oscillations with viruses and bacteria to accelerate their activity inhibition. Therefore, we proposed a compound method from surface characteristic and electric field and presented the AC and DC pulse electric stimulation with various frequency to investigate its inhibition on different virus and bacteria in this manuscript. The experimental indicated that the activity of virus and bacteria can be effectively inhibited in specific frequency. From the results, the suitable electric stimulation parameters for virus or bacteria were summarized and these information can provide the reference information for future application in environment virus activity inhibition and improving the public health. Please find the Conclusion in the revised paper.

Round 3
Reviewer 2 Report
Well done.